



# Impact of geographic variations of convective and dehydration center on stratospheric water vapor over the Asian monsoon region

K. Zhang[1], R. Fu[1], T. Wang[2*], Y. Liu[3]

[1] Jackson School of Geosciences, The University of Texas at Austin, Austin, Texas, United States.
[2] Department of Atmospheric Sciences, Texas A&M University, College Station, Texas, United States.
[3] State Key Laboratory of Numerical Modeling for Atmospheric Sciences and Geophysical Fluid Dynamics, Institute of Atmospheric Physics, Chinese Academy of Sciences, Beijing, China.
[*] Now at Jet Propulsion Laboratory, California Institute of Technology, Pasadena, California, United States.

*Correspondence to*: K. Zhang (kzkaizhang@utexas.edu)

**Abstract.** The Asian monsoon region is the most prominent moisture center of lower stratospheric (LS) water vapor during boreal summer. Previous studies have suggested that the transport of water vapor to the Asian monsoon LS is controlled by dehydration temperatures and convection mainly over the Bay of Bengal and Southeast Asia. However, there is a clear geographic variation of convection associated with the seasonal and intra-seasonal variations of the Asian monsoon circulation, and the relative influence of such a geographic variation of convection vs. the variation of local dehydration temperatures on water vapor transport is still not clear. Using the Aura Microwave Limb Sounder (MLS) satellite observations and a domain-filling forward trajectory model, we show that almost half of the seasonal water vapor increase in the Asian monsoon LS are attributable to the geographic variations of convection and resultant variations of dehydration center, comparable to the influence of the local dehydration temperature increase. In particular, dehydration temperatures are coldest over the southeast and warmest over the northwest within the Asian monsoon region. Although convective center is located over the southeastern Asia, an anomalous increase of convection over the northwestern Asian monsoon region increases the local diabatic heating in the tropopause layer and air mass entering the LS that is dehydrated at relatively warmer temperatures. The warmer dehydration temperatures allow anomalously moist air enters the LS and then moves eastward along the northern frank of the monsoon anticyclonic flow, leading to wet anomalies in the LS over the Asian monsoon region. Likewise, when convection increases over the southeastern Asian monsoon region, dry anomalies appear in the LS. On seasonal scale, this feature is associated with the march of the monsoon circulation, convection and diabatic heating towards the northwestern Asia monsoon from June to August, leading to an increasing fraction of the air mass to





be dehydrated at warmer temperatures over the northwestern Asian monsoon region. Work presented here confirms the dominant role of temperatures and also emphasizes that one should take the geographic variations of dehydration center into consideration when studying water vapor variations in the LS, as it is linked to changes of convection and large-scale circulation.

## 1   Introduction

Water vapor variation in the lower stratosphere (LS) contributes significantly to the global climate change through radiation budget (Forster et al., 1999; Solomon et al., 2010; Dessler et al., 2013) and chemical processes, particularly for ozone depletion (Evans et al., 1998; Dvortsov and Solomon, 2001; Shindell, 2001; Stenke and

Grewe, 2005; Anderson et al., 2012). Water vapor in the LS exhibits localized maximum over the Asian monsoon region from May to September (Rosenlof et al., 1997; Randel et al., 2001; Dessler and Sherwood, 2004; Milz et al., 2005; Park et al., 2007; Randel et al., 2015). This center of maximum water vapor is an important moist source for the global stratosphere (e.g., Randel et al., 2001; Gettelman et al., 2004; Ploeger et al., 2013), although its contribution relative to that of the tropical LS is still a subject of active research

(Rosenlof et al., 1997; Fueglistaler et al., 2005; Wright et al., 2011). The global models have suggested the importance of the Asian monsoon for water vapor transport to the tropics and global stratosphere (Dethof et al., 1999; Bannister et al., 2004; Gettelman et al., 2004; Ploeger et al., 2013). Therefore, it is important to understand the controlling transport processes of water vapor into the Asian monsoon LS.

The maximum water vapor concentration in the Asian summer monsoon LS is a result of convective transport of

moist air trapped by strong monsoon anticyclonic circulation (Dunkerton, 1995; Jackson et al., 1998; Dethof et al., 1999; Park et al., 2007). However, its transport pathways are still debatable. Most of convection within the Asian monsoon region only reaches about 200 hPa (~12.5 km above the sea-level) (e.g., Fu et al., 2006; Park et al., 2007; Wright et al., 2011). Moist air transported by convection at this level would be dehydrated as it slowly ascent to the tropopause (Holton and Gettelman, 2001). Consequently, the occurrence of deep convection is not

significantly correlated with the variation of LS water vapor within the Asian monsoon region (Park et al., 2007).

The relative impact of convection and temperatures on LS water vapor variations is still an undergoing study. Fu et al. (2006) suggested that deep convection over the Tibetan Plateau and south slope of the Himalayas can reach the tropopause more frequently. Together with warmer tropopause temperatures, they can be the main source of water vapor for the LS over the Asian monsoon region. Some studies suggest that over the Bay of



Bengal and Southeast Asia water vapor can be transported into the LS via direct convective injection (Park et al., 2007; James et al., 2008; Devasthale and Fueglistaler, 2010). Wright et al. (2011) have compared the relative contributions of three distinct convective regimes within Southeast Asia (i.e., South Asian subcontinent, the South China and Philippine Seas, and the Tibetan Plateau and South Slope of the Himalayas) to the seasonal

variation of water vapor in the tropical lower stratosphere. They found large discrepancies among the three most commonly used reanalysis products.

Recently, Randel et al. (2015) shows that there is a dominant impact of temperatures over the southeastern flank of the Asian anticyclone on the intra-seasonal variations of water vapor over the Asian monsoon regions, and overshooting deep convection plays a relatively minor role. However, the seasonal temperature changes in this

region cannot fully explain the continuous increase of water vapor in the Asian monsoon region during summertime, which will be shown in this paper. Besides, there is a clear geographic variation of convection associated with the seasonal and intra-seasonal variations of the Asian monsoon circulation, and the relative influence of such geographic variation of convection vs. the variation of local dehydration temperatures on water vapor transport is still not clear. In this study, we aim to clarify this question and the role of convection by

analyzing water vapor transport based on the Aura Microwave Limb Sounder (MLS) daily observations and a domain-filling forward trajectory model.

## 2    Data and Methodology

In order to examine the relationships between water vapor, temperatures and convection over the summertime

monsoon region, we use water vapor observations from the Aura MLS and outgoing longwave radiation (OLR) from NOAA (National Oceanic and Atmospheric Administration) in 2005-2013 boreal summers (May – September). We grid level 2 version 3.3 daily MLS water vapor to 10°x5° longitude by latitude from 215 hPa to 100 hPa. Within this vertical range, the MLS $H_2O$ precisions are 0.5–0.9 ppmv (Livesey et al., 2011). NOAA interpolated daily OLR in 2.5°x2.5° horizontal resolution is used as a proxy for convection. The diabatic heating

data used for analysis is from the European Centre for Medium-Range Weather Forecasts (ECMWF) Interim Re-Analysis (ERA-Interim) archive (output from the reanalysis model forecast fields) (Dee et al., 2011). Singular value decomposition (SVD) (Bretherton et al., 1992) is applied to the covariance matrix between daily anomalies of water vapor and OLR to objectively identify the coupled patterns and the maximum covariance.



To explain the observational relationships, we use a domain-filling forward trajectory model to identify the water vapor transport pathways over the summertime Asian monsoon region. The trajectory model used here follows the details described in Schoeberl and Dessler (2011), with trajectory integration calculated from Bowman trajectory code (Bowman, 1993; Bowman et al., 2013). Two trajectory runs driven by different circulation and temperature fields from two reanalyses are compared to evaluate the uncertainties: (1) using ERA-Interim fields (Dee et al., 2011), denoted as traj_ERAi; (2) using Modern Era Retrospective Analysis for Research and Applications (MERRA) fields (Rienecker et al., 2011), denoted as traj_MERRA. Previous studies have shown that this model is able to simulate both water vapor (Schoeberl and Dessler, 2011; Schoeberl et al., 2012; Schoeberl et al., 2013; Wang et al., 2015) and chemical tracers (Wang et al., 2014) in the LS very well. In this study, we analyze results from diabatic run in isentropic coordinates, in which the potential temperature tendency is converted from the diabatic heating rates as vertical velocity. Parcels are initialized at tropical 370K isentropic level, which is above the level of zero radiative heating (~355-365K) (Gettelman et al., 2002) but below the tropical tropopause (~375-380K). Along the trajectory integration we use 100% saturation level with respect to ice to remove excess of water vapor instantly, i.e. the "instant dehydration", which has been proven effective in simulating stratospheric water vapor (e.g., Fueglistaler et al., 2005; Schoeberl et al., 2014). The term "last dehydration" is used in this study to indicate the latest dehydration event along the historical travel path of an air parcel. Details of parcel initialization and removal criterion for water vapor along the trajectories can be referred to Wang et al. (2015). All outputs in the trajectory model have been weighted by the MLS average kernels for fair comparison.

## 3 Results

### 3.1 Seasonal enhancement of LS water vapor over the Asian monsoon region

During boreal summer, there is an isolated moisture center observed over the Asian monsoon region (Rosenlof et al., 1997; Randel et al., 2001; Dessler and Sherwood, 2004; Milz et al., 2005; Park et al., 2007; Randel et al., 2015). It is featured with LS water vapor increases throughout the summertime from May to August (e.g., Randel et al., 2015), as shown by the black line in Figure 1. The similar temporal variation, although with dry biases, can also be simulated by the trajectory model (blue and green lines in Fig. 1), in which water vapor is determined by temperatures at last dehydration locations. Since the trajectory model has larger uncertainties in simulating water vapor below 100 hPa, we applied weighting functions only to levels above 100 hPa, which makes air slightly drier due to less convective influences from air below. The agreement between the trajectory



model simulation and observed water vapor variation implies that temperatures, rather than convective injection, dominates the observed seasonal enhancement over the Asian monsoon region. Both observations and trajectory model calculations of the Asian monsoon have shown that dehydration primarily occurs on the cold (equatorward) side of the LS anticyclonic circulation (Wright et al., 2011; Randel et al., 2015). However, the

seasonal changes of 100 hPa temperatures over the southeastern flank of the anticyclone (15-32$^{\circ}$N, 70-120$^{\circ}$E, the red line in Fig. 1), which are expected to dominate dehydration of the LS over the Asian monsoon region (Randel et al., 2015), do not show as significant increase as water vapor from May and June to August. Thus, temperatures at this location cannot fully explain the increase of water vapor in the LS of the Asian monsoon region from early to late summer.

To identify the temperature controlling areas for each month, we use the trajectory model to determine the last dehydration locations for all the air parcels at 100 hPa over the Asian monsoon region (20-40$^{\circ}$N, 40-140$^{\circ}$E). Figure 2 shows the probability density distribution of last dehydration locations during May to August (white contours) and averaged tropopause temperatures (color shadings). A common feature is that dehydration mostly occurs on the cold (equatorward) side of the Asian monsoon anticyclonic circulation, consistent with Wright et

al. (2011). This behavior of dehydration through the cold temperatures is similar to the freeze-drying process near the cold-point tropopause over the western Pacific (Holton et al., 1995; Holton and Gettelman, 2001). During May, most of the air parcels are dehydrated over the southeastern Asia, where temperatures are lower than other Asian monsoon areas. The center of dehydration shifts north-westward to the Bay of Bengal in June (Fig. 2b), then expands to the Bay of Bengal/north India in July (Fig. 2c) and finally shifts to the north India

during August (Fig. 2d). A statistical view of the westward shift of dehydration is shown in Figure 3. From early to late summer, the fraction of air parcels that are dehydrated over the west side of the Asian monsoon region (red line) is increasing gradually, opposite to the east side (blue line). During September, the dehydration starts to retreat from the west side towards the east side.

In order to quantify the relative impact of geographic variations of dehydration center and local temperature

changes on water vapor increase from June to August respectively, we conducted three idealized experiments based on traj_ERAi and traj_MERRA (Table 1). Taking traj_ERAi as an example, CTL is a control experiment with dehydration pattern (i.e. last dehydration locations) and temperatures set the same as in June. When temperatures are changed to temperatures in August (Exp_TEM) and everything else is unchanged, the averaged water vapor in the Asian monsoon LS is increased by on average ~0.49 ppmv (standard deviation is σ= 0.21

ppmv), suggesting a moistening effect of the local dehydration temperature changes from June to August.



Similarly, when dehydration pattern is modified by replacing the dehydration locations for all air parcels during June by those in August, while keeping the dehydration temperatures unchanged from those of June (Exp_LOC), the averaged water vapor is increased by on average ~ 0.59 ppmv (σ= 0.24 ppmv), also showing a moistening effect of the geographic shift of dehydration center from June to August. Thus, the westward geographic shift of

dehydration center toward warmer temperatures over the western Asian monsoon region could contribute to a significant increase of the total LS water vapor from June to August, comparable to the contribution of the local temperature changes. Results from the traj_MERRA are consistent with traj_ERAi, even with some minor differences (Table 1). But both trajectory runs indicate that the westward shift of dehydration significantly enhances water vapor in the Asian monsoon LS from early to late summer.

The diabatic heating, which drives the vertical transport in the trajectory model, also shows a westward shift with stronger enhancement of rising motion over the west side during August compared to that during May and June (Fig. 4). This suggests a westward migration of vertical transport associated with gradual increase of convection over northwestern and gradual decrease of convection over the southeastern Asian monsoon region from early to late summer. Such an anomalous shift of convection and diabatic heating near the tropopause

towards northwest side is a common feature over the Asian monsoon region during the boreal summer (e.g., Qian and Lee, 2000), leading to increasing fraction of air mass in the LS coming from the warmer west part than the colder east part of Asian monsoon region.

### 3.2 Intra-seasonal variations of LS water vapor over the Asian monsoon region

To evaluate the relationships between the lower stratospheric water vapor variation and convection and diabatic

heating objectively, we applied SVD analysis described in Section 2 to the data. Figure 5 shows the dominant mode of SVD for MLS water vapor anomalies in the UT/LS and OLR anomalies. The pattern of water vapor anomalies at 100 hPa in Figure 5a shows uniform anomalies over the entire Asian monsoon region with largest values over the western region. The corresponding OLR anomaly pattern (Fig. 5d) illustrates a zonal east-west dipole pattern, i.e., anomalously strong convection in the southwest and anomalously weak convection in the

southeast of the Asian monsoon region, respectively. The statistics of the first SVD mode confirm that water vapor anomalies at 100 hPa over the Asian monsoon region during summer are significantly correlated with OLR anomalies (r = 0.41, p<0.01). This first SVD mode accounts for a significant 42.63% of the total water vapor variance. By regressing 147 hPa water vapor anomalies onto the principle component of the OLR SVD mode, we obtain the heterogeneous map of water vapor anomalies at 147 hPa (Fig. 5b). There is anomalously

higher humidity in the entire UT over the southwestern Asian monsoon region, where an increase of cirrus




clouds occurs (not shown), associated with a westward shift of convection. The heterogeneous maps of diabatic heating and dehydration frequency anomalies are consistent with the westward shift of convection, suggesting the latter probably enhances ascending motion and hence fraction of air mass being dehydration over the western Asian monsoon region (Figs. 5c-d). The consistent wet anomalies from the UT to the LS, along with

enhanced diabatic heating and dehydration over the western Asian monsoon region, suggest that the increase of LS water vapor over the Asian monsoon region is associated with enhanced water vapor transport driven by anomalous convective diabatic heating over the western monsoon region.

To further look at the intra-seasonal oscillation of dehydration center associated with LS water vapor changes, we compared the behavior of last dehydration locations during wet and dry days (Fig. 6). We use the trajectory

runs to identify the last dehydration locations for all the air parcels at 100 hPa in the Asian monsoon region during wet and dry days, respectively, and compare their differences. We first evaluate the performance of the model simulation on 100 hPa water vapor variations over the Asian monsoon region (20-40ºN, 40-140ºE). Figures 6a-b show the time series of 100 hPa water vapor anomalies in the Asian monsoon region during summer from the trajectory runs (red line) compared with the Aura MLS observations (black line, same as

Figure 4a in Randel et al. (2015)). The trajectory model can simulate both interannual and intra-seasonal variations of water vapor in the Asian monsoon very well (r = 0.6 for traj_ERAi and r = 0.51 for traj_MERRA), including most wet and dry extremes, which also further confirms the dominant role of temperatures on controlling the LS water vapor variations. The traj_MERRA has a bad performance during 2006 compared to traj_ERAi, leading to the relatively low correlation with observations. By selecting the wet and dry events

(above and below one standard deviation, denoted by the red and blue dashed lines) in the trajectory runs, we further investigate the different dehydration and initialization behavior during different moist states in the LS. As shown in Figures 6c-d, during wet events, there is more dehydration over the western Asian monsoon region and less over the eastern region compared with dry events, implying a westward shift of dehydration. The underlying initialization behavior at 370K also shows a westward shift of air source with more air from the west

side and less air from the east side during wet events than dry events (Figs. 6e-f). Such a clear westward shift of initialization and dehydration activities during wet events is consistent with the observational results of enhanced water vapor transport over the west side shown in Figure 5. This west-east oscillation of vertical transport is associated with changes of the large-scale circulation forced by diabatic heating in the trajectory model. These results are consistent between traj_ERAi and traj_MERRA, further confirming the robust physical

links between convection, dehydration and water vapor transport. It is important to consider the west-east





oscillations of anomalous convective activity and dehydration center, in order to fully explain the changes of LS water vapor at both seasonal and intra-seasonal time scales.

## 4    Conclusions and Discussion

Water vapor in the Asian monsoon LS significantly influences water vapor of the global stratosphere (Dethof et
al., 1999; Bannister et al., 2004; Gettelman et al., 2004; Ploeger et al., 2013), but significant uncertainty still exist as to what processes that control its transport. This paper clarifies the impact of dehydration locations – especially its geographic variations – on water vapor transport to the LS through a joint analysis of satellite data and trajectory model simulations. Although our focus is at 100 hPa, we have also examined the results for water vapor variations at 82 hPa and found similar results.

First, our result confirms the dominant role of temperatures on LS water vapor variations (Fig. 1 and Fig. 6a). This is consistent with the study of Randel et al. (2015). However, this study suggests that aside from the temperatures changes, the variation of the dehydration locations, especially its west-east oscillation, also plays a significant role in both the intra-seasonal and early-to-summer moistening of the LS over the Asian monsoon region. In particular, the dehydration locations vary with time, characterized by a westward migration from the
southeastern Asian monsoon region with colder temperatures in May to northwestern India with warmer temperatures in August (Fig. 2). The westward geographic shift of the dehydration center to warmer temperatures over the western Asian monsoon region could increase water vapor significantly, which is comparable to the influence of local temperature changes (Table 1). This sub-seasonal migration of dehydration is associated with a westward migration of vertical motion as shown in Figure 4, corresponding to the seasonal
march of the Asian monsoon convective systems (e.g., Qian and Lee, 2000). Second, we confirm the physical link between convection, diabatic heating and large-scale transport. At an intra-seasonal time scale, the westward shift of convection appears to enhance the diabatic heating and the ascending motion in the tropopause layer, which in turn enhances the vertical transport of water vapor and dehydration frequency over the west side of the Asian monsoon region. The less dehydrated air ascends and transports eastward, eventually
increasing LS water vapor over the entire Asian monsoon region (Fig. 5). More transport and dehydration under warmer tropopause temperatures in the west of the Asian monsoon region would contribute to the wet anomalies in the LS (Fig. 6). The east-west oscillation of convection and dehydration patterns is similar to the well-known Boreal Summer Intraseasonal Oscillation in the Asian monsoon region (e.g., Lau and Chan, 1986; Lawrence and Webster, 2002; Kikuchi et al., 2012), suggesting a link between large-scale monsoon circulation with water
vapor transport into the LS. In summary, by confirming the dominant role of temperatures on water vapor



variations in the LS, we emphasize the importance of the role of convection on temperature changes and the geographic locations of dehydration.

Most of the moist air is from the dehydration over the western Asian monsoon region where the temperatures are warmer than the cold-point temperatures. However, the source region of the moist air remains unclear. The

Bay of Bengal and Southeast Asia have previously been identified as primary source regions of air parcels over the Asian monsoon LS due to convective protrusion (Park et al., 2007; James et al., 2008; Devasthale and Fueglistaler, 2010). Some of the air from this region detrains in the UT and is advected south-westward (e.g., Park et al., 2009), possibly contributing to the moist air entering the LS over the western Asian monsoon region shown in this study. Meanwhile, some of the air entering the LS over the Bay of Bengal and Southeast Asia, is

dry mostly due to the substantial dehydration by cold-point temperatures.

This studies implies that geographic changes in convection patterns within the South Asian monsoon region could change the abundance of water vapor in the LS of the Asian monsoon without changing the strength of convection. For example, a westward convection shift would likely enhance water vapor transport through a warmer pathway from the UT to the LS over the west side. Some studies indicate long-term changes in

precipitation over the western and surrounding regions in the past decades (e.g., Ramanathan et al., 2005; Goswami et al., 2006; Gautam et al., 2009; Bollasina et al., 2011; Turner and Annamalai, 2012; Zuo et al., 2013; Walker et al., 2015). However these changes are not clearly evident due to large discrepancies between different datasets (Walker et al., 2015). How these changes influence LS water vapor is also unknown. Therefore, studies of convection regime variation will have important implications for predicting future stratospheric water vapor

changes in the Asian monsoon regions, and possibly over the globe.

*Acknowledgement.* We sincerely thank Bill Randel and Mejeong Park for their comments and discussions that lead to significant improvement of this work. We thank Peirong Lin for improvement on figures and writing. We would like to acknowledge the editorial assistance from Adam Papendieck. K. Zhang and R. Fu were supported by NASA Aura Science

Team Grant (No. NNX11AE72G). Y. Liu is supported by National Science Fundation of China (NSFC 91437219). MLS data were obtained from the MLS web site (http://mls.jpl.nasa.gov/products/h2o_product.php), the gridded OLR data from NOAA-CIRES Climate Diagnosis Center (http://www.cdc.noaa.gov/). The ERA-Interim data were downloaded from http://apps.ecmwf.int/datasets/data/interim-full-daily/.



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



Table 1. Three idealized experiments based on traj_ERAi (values in bold) and traj_MERRA (values in brackets) to quantify the relative contributions of dehydration shift and dehydration temperatures on averaged water vapor over the Asian monsoon region.

| Experiments | Dehydration pattern | Dehydration temperatures | Averaged water vapor (unit: ppmv) |
|---|---|---|---|
| CTL | June | June | **3.71** (3.80) |
| Exp_TEM | June | August | **4.20** (4.16) |
| Exp_LOC | August | June | **4.30** (4.22) |



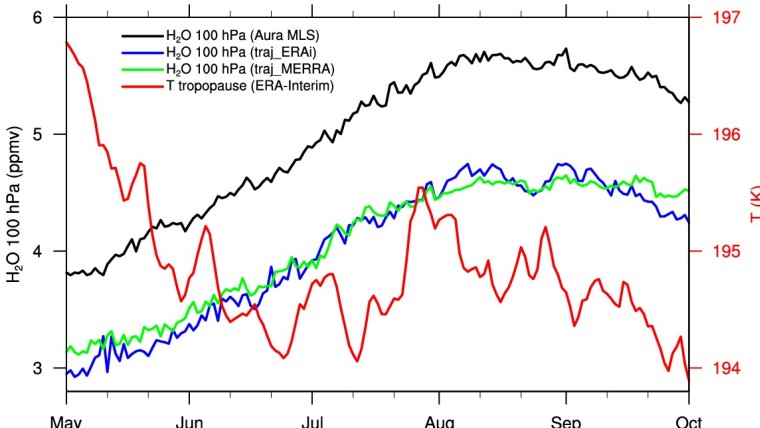

**Figure 1.** Climatological 100 hPa water vapor observed by the Aura MLS (black) and simulated by the traj_ERAi (blue) and traj_MERRA (green). The time series were averaged over the Asian monsoon region (20-40ºN, 40-140ºE) during May-September, 2005-2013. The red line denotes the time series of tropopause temperatures averaged over the same domain as in R15 (15-32ºN, 70-120ºE).



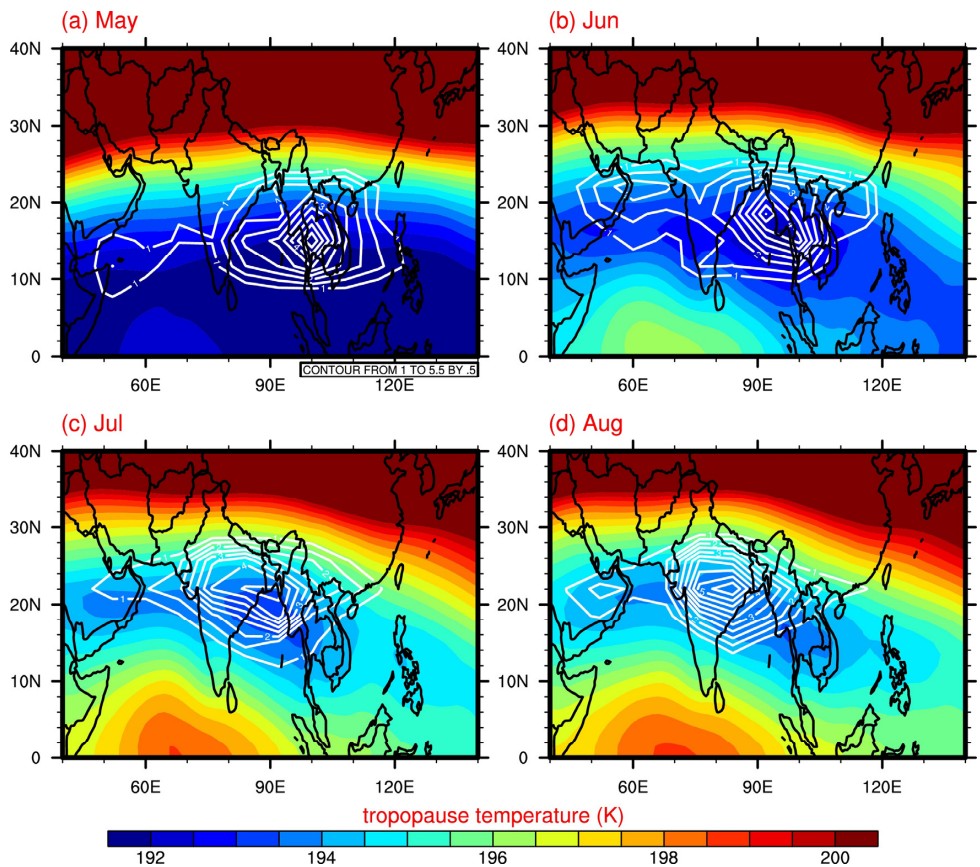

**Figure 2.** Probability density distributions of last dehydration locations for air parcels located at 100 hPa over the Asian monsoon region during May to August derived from traj_ERAi (white contours from 1% to 5.5% with an interval of 0.5%). Color shadings are averaged tropopause temperatures in ERA-Interim.



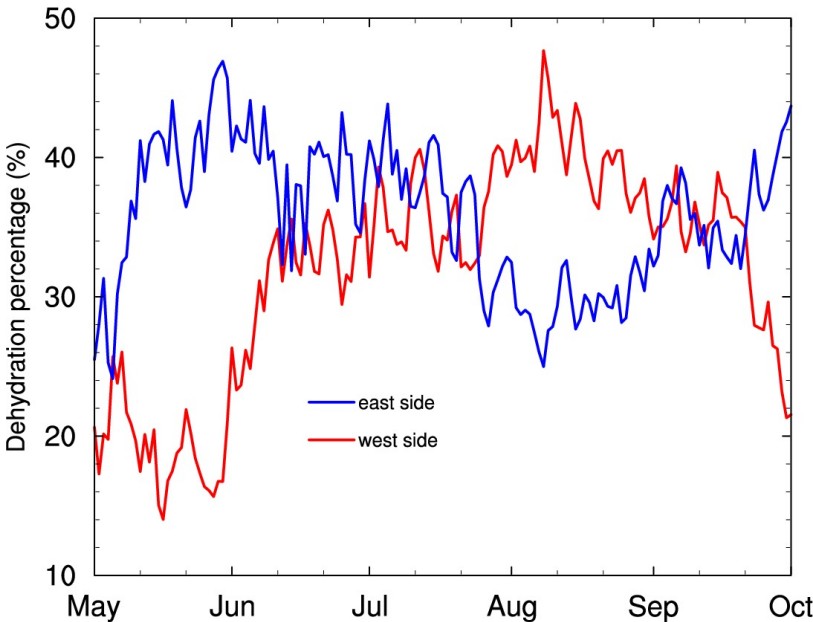

**Figure 3.** Climatological time series of percentage of dehydration (%) located over the west side (15-35°N, 40-80°E, red line) and the east side (10-30°N, 90-120°E, blue line) for air parcels located at 100 hPa over the Asian monsoon region based on traj_ERAi.



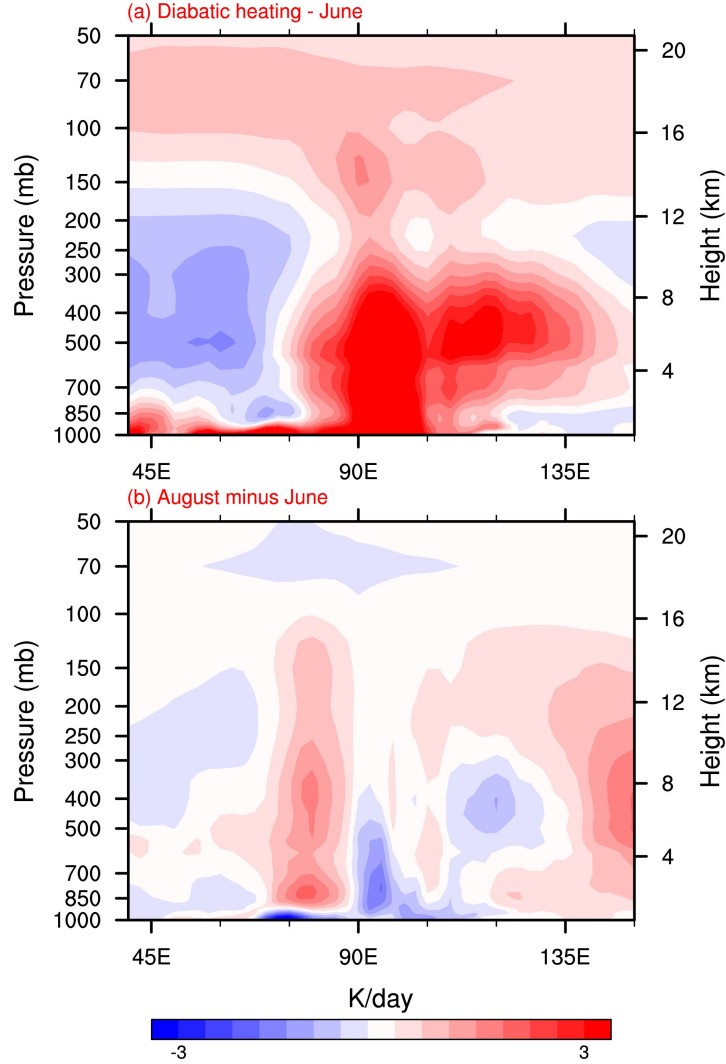

**Figure 4.** Vertical-longitude distribution of climatological diabatic heating averaged within 15-30°N in ERA-Interim for (a) June; (b) difference between August and June.





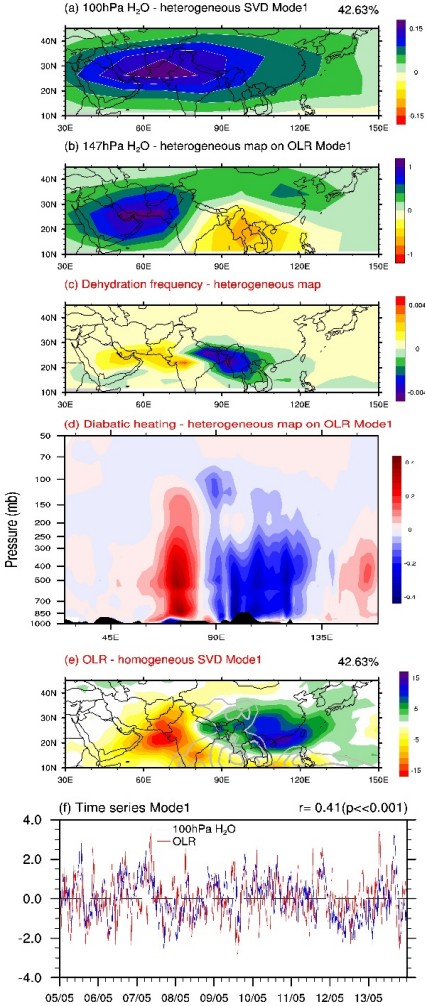

**Figure 5**. First SVD mode between water vapor anomalies (ppmv) and OLR anomalies (W/m²) over the Asian monsoon region. (a-b) Heterogeneous map of water vapor at 100 hPa and 147 hPa on OLR SVD mode 1; (c) Heterogeneous map of dehydration frequency derived from traj_ERAi on OLR SVD mode 1; (d) Heterogeneous map of diabatic heating at 25ºN on OLR SVD mode 1; (e) homogeneous map of OLR SVD mode 1; (f) standardized time series of 100 hPa water vapor anomalies and OLR anomalies.



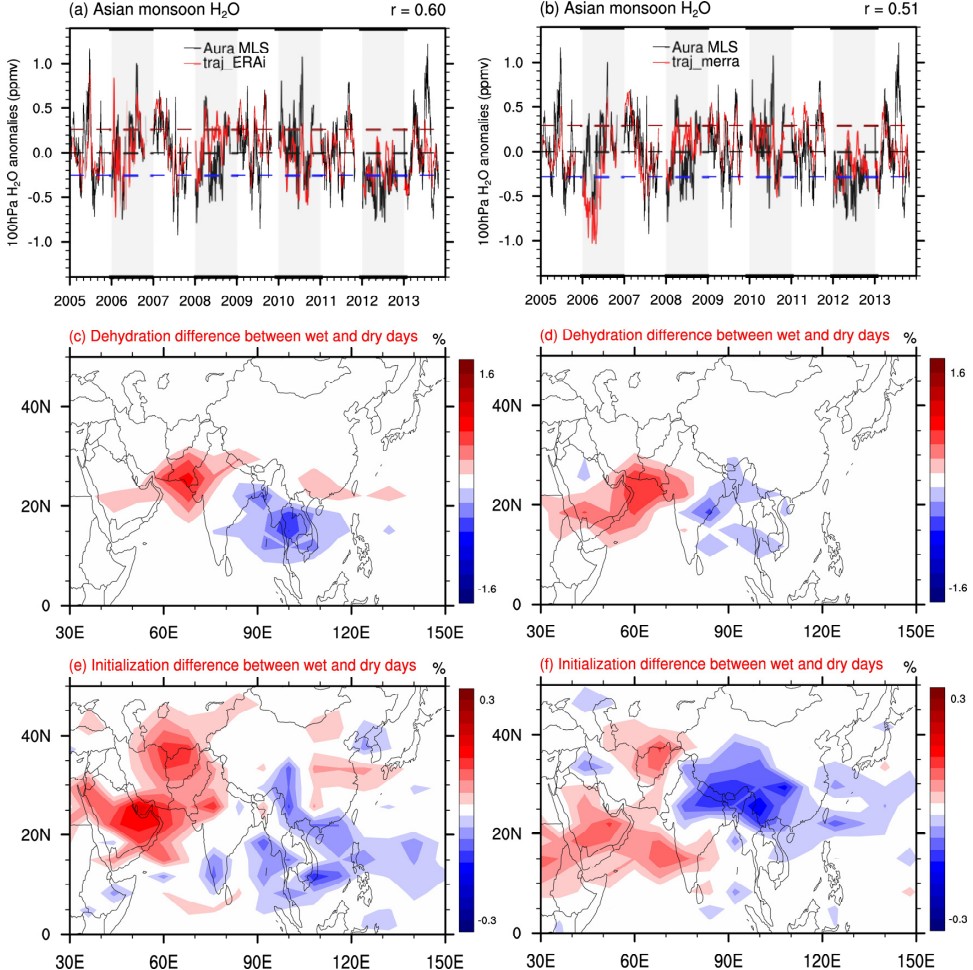

**Figure 6.** Left panel is from traj_ERAi, right panel is from traj_MERRA. (a-b) Time series of deseasonalized 100 hPa water vapor anomalies in the Asian monsoon region (20-40ºN, 40-140ºE) derived from Aura MLS observations (black line) and trajectory runs (red line). Dashed lines are at one sigma variability for water vapor time series in trajectory runs, and are used to identify wet and dry event. (c-d) Anomalous frequency of occurrence of the last dehydration locations between wet and dry days for air parcels located at 100 hPa over the Asian monsoon region. (e-f) are same as (c-d) but for initialized locations.