# Peer review of "Impact of geographic variations of the convective and dehydration center on stratospheric water vapor over the Asian monsoon region"

_Atmospheric Chemistry and Physics, 2016_

## Referee Comment (RC1) · Anonymous Referee #1 · 15 Feb 2016

Review of Atmos. Chem. Phys. Discuss., doi:10.5194/acp-2016-21, 2016
Title: Impact of geographic variations of convective and dehydration center on stratospheric water vapor over the Asian monsoon region

**General comments:**

This paper discussed the contribution of the geographic variation of convection and the associated geographic variation of dehydration locations to the water vapor

over Asian monsoon region in the lower stratosphere during boreal summer. The trajectory model simulation provided clear proof of the east-to-west shift of the dehydration location at the intra-seasonal time scale. Further SVD analysis confirms the connection between the convection pattern and the water vapor anomalies.

The main concern from my perspective is the statements in many places of the paper like 'warmer tropopause temperature in the west of Asian monsoon region', e.g. Page 1 line 20-24, Page 6 line 16-17 and Page 8 Line 25-26. It seems to me that the author is comparing the tropopause temperature in the west side to that in the east side. However, it is not clear in which region, which latitude or which period it is compared.

The author divided the east side and west side of Asian monsoon region by 80-90°E according to the caption of Figure 3. However, from the tropopause temperature shown in Figure 2, the differences of the tropopause temperature between the east side and the west side are not significant. And the author also pointed out that the convection increases over the west side of Asian monsoon region which increases the local diabatic heating. From my understanding, the anomalous convection over the west Asian monsoon region should lead to stronger upwelling and relatively cold temperature in the tropopause layer, which controls the dehydration. Actually, the pattern of dehydration location and of the tropopause temperature is associated with each other. That means when the convection increases over western side of Asian monsoon region, the cold point temperature should correspondingly decreases to some degree. I suggest author gives a direct comparison of the tropopause temperature of east side and west side to clarify this point.

Furthermore, Figure 1 shows the variation of tropopause temperature is not able to fully explain the variation of seasonal water vapor variation, which is the motivation of the work. I have some questions concerning the domain and the magnitude of interannual variabilities, which are specified in the following part.
The paper describes interesting result, which contributes to complete the picture of moisture center over Asian summer monsoon region. Overall, the paper is nicely structured and presented. I suggest it is published after clarifying the questions above.

**Specific comments:**

- *Pg. 2, line 21: A recent paper (Ploeger et al. 2015, ACP) intensively discussed the variability of a PV-based transport barrier of Asian monsoon anticyclone. This study is highly related here and I recommend to cite this study. doi:10.5194/acp-15-13145-2015*

- *Pg. 3 , line 28: I suppose the OLR data is also daily anomalies according to the section 3.2. So add 'of' after 'water vapor and' in order to avoid the misunderstanding.*

- *Pg. 4, line 29: Is the weighting functions the weighting matrix of MLS averaging kernels? If yes, please clarify here.*

- *Figure 1: First, this figure shows the 9-year climatology of water vapor and tropopause temperature. The intra-seasonal variations are usually can be 'offset' by averaging over several years. Can you comment on how large is the interannual variability of temperature? Does this strong intra-seasonal variations of temperature attribute to some particular year or is it a common feature for this domain during boreal summer? Perhaps it worth to add the standard deviation to the tropopause temperature.*

- *Figure 1: Second, you mention that the same domain used for area-averaging the tropopause temperatures as R15, which is 15-32° N, 70-120° E. However, I checked R15 and the domain 15-30° N, 70-120° E is used. Besides, you also use 15-30° N in Figure 4 of averaging the diabatic heating rate. From Figure 2, it is seen that the gradient of tropopause temperature around 30° N is very large and the 2 degrees could influence the variation of temperature shown in this plot. I suggest to show the tropopause temperature averaged over 15-30° N, 70-120° E. Otherwise the author could compare the results between the 2 domains and clarify the results stays the same.*

- *Figure 2 and 3: I suggest to add boxes in Figure 2 to show the domains of west side and east side mentioned in the caption of Figure 3.*

- *Figure 5: the subfigures are too small. It is better to enlarge the figure, especially those color bars.*

- *Pg. 8, line 10: it should be '(Fig.1 and Fig.6a-b)'*

---

## Referee Comment (RC2) · Anonymous Referee #2 · 27 Mar 2016

First, my apologies to the authors and the editor for the long delay in publishing this review.

This manuscript examines how changes in the distribution of convective sources and dehydration locations of air in the Asian monsoon upper tropospheric anticyclone affect the amount of water vapour entering the lower stratosphere in this region. The concept is worthwhile, and the paper makes some valid points about how seasonal and intraseasonal variability in convective sources influence the moisture content of air near the tropopause. However, some aspects of the methodology and argument are problematic, and require more justification at the very least.

The biggest weakness of this paper is that it takes water vapour variations at 100 hPa

to be representative of the 'lower stratosphere', neglecting previous work indicating that final dehydration for air entering the tropical stratosphere via this region typically occurs at lower pressures / higher altitudes. This does not necessarily invalidate the core conclusions of this paper (the processes controlling water vapour variability at 100 hPa are also important to understand, particularly if they propagate to higher levels), but it does imply strong limitations on their applicability that are not effectively communicated or explored in the paper. At the very least, the authors should clarify that 'LS' in this case means 100 hPa, and discuss the limitations that that entails. Even better, the authors could use their trajectory simulations to connect the results and conclusions at 100 hPa to final dehydration statistics and stratospheric entry mixing ratio. In other words: do these intraseasonal differences in source location / temperature distribution / transport affect the amount of water vapour entering the global stratosphere via the Asian monsoon anticyclone, and, if so, how much? These additions would help tremendously in establishing how this work fits in the context of other studies of water vapour transport and variability in this region.

**1    General comments**

1. My main concern is that the analysis focuses almost exclusively on water vapour at 100 hPa, and particularly that this is assumed to represent lower stratospheric water vapour. The vertical location of dehydration for air entering the stratosphere varies quite a lot, and is typically higher (in altitude) than 100 hPa. Would the results still be valid for variations in water vapour at 83 or 68 hPa, or are they only relevant to a shallow layer bracketing the tropopause? If I have understood the analysis correctly, this might be checked by analyzing the 'final dehydration' locations and temperatures for these trajectories during transit to the stratosphere in addition to the 'latest dehydration' locations and temperatures for the model results at 100 hPa. Are the statistics of final dehydration for these trajectories

significantly different from those with convective sources over the eastern side of the region? Regardless, more needs to be done here, either to connect these results more clearly to stratospheric entry mixing ratio (post-final dehydration) or clearly distinguish between studies for which 'LS' means 'above the tropopause layer' and/or 'after final dehydration' and this study (where 'LS' means 100 hPa, well within the tropopause layer and likely prior to final dehydration).

2. I'm not convinced that the idealized experiments that separate the dehydration temperatures and dehydration locations are viable in this case. This approach works well when either temperature changes or circulation changes are dominant (and therefore separable), but has little meaning when temperature and circulation changes are tightly coupled. Another way of thinking about this is that separability is a justifiable assumption in situations where changes in dehydration location are dominated either by (1) an unchanged circulation sampling a modified temperature distribution or (2) a modified circulation sampling an unchanged temperature distribution. Too much overlap between these situations results in degeneracy, at which point the contributions of temperature changes and circulation changes cannot be reliably distinguished. My expectation is that in this case the tight couplings among convection, circulation (especially diabatic heating) and temperature at 100 hPa violate separability, as also briefly mentioned by reviewer #1. I am willing to be convinced otherwise, but additional justification for these simulations is needed if they are to be used as supporting evidence here.

3. This work would benefit from a more complete analysis of confidence intervals and significance testing. As also noted by reviewer #1, many of the arguments rely on changes and/or differences that are relatively small. This is particularly relevant for the time series in Fig. 3 and the August minus June differences shown in Fig. 4 and Table 1, and otherwise reported in the text.

4. The text is clear for the most part, but the manuscript would benefit from Englishlanguage editing by a colleague or professional editor. There are a few points
where editing will be necessary to improve the clarity; see technical comments
below.

**2 Specific comments**

**p.2, l.24-25:** this sentence is vague and should be reworded for clarity — at seasonal
time scales and large spatial scales both the temporal evolution and geographic dis-
tribution of LS water vapour are correlated with convective activity, it's just that these
correlations do not generally extend to variability within the anticyclone itself.

**p.2, l.28:** Given the uncertainties and competing hypotheses put forward by subse-
quent studies, it would be more appropriate to change 'they can' to 'they proposed that
this convection can'

**p.3, l.6:** Wright et al. (2011) did find large discrepancies among the different reanalysis
data sets, but the qualitative results were robust: trajectories originating from convec-
tion over Tibet were consistently moister but less numerous than trajectories originating
from convection over the other regions, so that these trajectories had relatively limited
impacts on water vapour in the global tropical LS.

**p.3, l.23:** It would be useful to note also the relative precision here (as xx–yy%)

**p.4, l.16:** In this case, since the focus is on the evolution at 100 hPa, I presume that
'latest dehydration' refers to most recent dehydration rather than final dehydration. This
choice should be stated explicitly to prevent confusion – 'latest dehydration' is by itself
too vague, as it could mean either 'most recent' or 'final'.

**p.4, l.18-19:** How is this done, by gridding the simulated water vapour mixing ratios
and then applying the averaging kernels to construct a vertical grid? I recommend
expanding slightly on this description. Also, as noted later on this page, the exclusion

of simulated values below 100 hPa results in a dry bias. I assume this statement is based on testing the sensitivity to whether those values are included. Does this testing indicate whether including/excluding the simulated values at lower levels has any impact on the qualitative evolution of the variability?

**p.5, l.5-9:** It is very difficult for the reader to evaluate the statement that '100 hPa temperatures over the southeastern flank of the anticyclone ... do not show as significant increase as water vapour from May and June to August'. This argument should be made more quantitative. This could be as simple as a calculation relating the May/June to August mean temperature change to a fractional change in mean saturation specific humidity (with appropriate uncertainty estimates), which can then be compared to the fractional change in simulated water vapour mixing ratio at 100 hPa (with appropriate uncertainty estimates).

**p.5, l.13:** (Fig. 2) Does 'tropopause temperatures' mean '100 hPa temperatures', or are these evaluated at a diagnosed tropopause?

**p.5, l.20:** (Fig. 3) Here it would be helpful to include also the evolution of mean 'latest dehydration' temperatures over the eastern and western parts of the domain, with uncertainty estimates. This would help to clarify that it is in fact the shift in dehydration location (and not the temperature evolution) that dominates the seasonal evolution of water vapour at 100 hPa, and could perhaps supplement or replace the idealized simulations in the overall argument.

**p.5, l.26:** (Table 1) If using these simulations, it would be useful also to include the August–August results to give a quantitative benchmark for evaluating the idealized June–August and August–June simulations. I know that these are shown in Fig. 1, but so are the June–June results. I could not find this number reported anywhere in the text.

**p.6, l.12:** (Fig. 4) Is there any benefit to including the profiles of diabatic heating below 300 hPa? Including these estimates requires the use of a relatively large scale,

and makes it difficult to distinguish the variations in the UTLS (which is what we are particularly interested in). Moreover, there is a negative anomaly centered around 70 hPa above the location with enhanced convective activity in August relative to June. Is this negative anomaly significant? If so, this suggests that upward motion in this region is weaker than during June above 100 hPa, which might mean that the trajectories involved circuit the anticyclone more times during ascent. This relates to general comment #1 above: how much does this westward shift of convective source location ultimately impact stratospheric entry mixing ratios?

**p.6, l.20:** (Fig. 5) The use of colours here is confusing, with red sometimes meaning a positive change and sometimes a negative change. Moreover, 'physically consistent with a moist anomaly in the UT' is sometimes indicated by the yellow/orange/red half of the colour scale and sometimes by the green/blue/purple half of the color scale. I recommend that you either make the use of this default colour table logically consistent across panels or use different colour tables for anomalies in different quantities. I also agree with reviewer #1 that the fonts are too small and difficult to read in several of the panels included in Fig. 5, and I can barely even see the variations of the lines in panel (f). If the variations for all of the years are necessary, perhaps it would be better to make panel (f) a separate figure and split it into multiple panels, one for each year? If not, it might be best to show the variations for a selected time period covering one or two years, so that those variations are easier to identify in the figure.

**p.7, l.1:** Which data is used to identify the increase in cirrus clouds?

**p.7, l.3:** What mechanism drives the enhanced ascending motion? Enhanced latent heating above 370 K? Enhanced cloud radiative heating? Is this enhanced ascending motion consistent between ERA-Interim and MERRA? This could be explored by looking at the components of the heat budget — both ERA-Interim and MERRA provide clear-sky and all-sky radiative heating products.

**p.7, l.9:** (Fig. 6) For clarity, the definitions of 'wet' and 'dry' days should perhaps be

moved from l.20 to here.

**p.7, l.18-19:** It would be useful to include the correlation for traj_MERRA if data from the mismatched period in 2006 is excluded.

**p.8, l.25:** The presented work only supports this statement if we consider 100 hPa to be representative of the LS in this region — no confirmation has been shown that this seasonal evolution in the convective source extends to lower pressures / higher altitudes, which should also be considered part of the LS.

**p.9, l.4:** What is meant by 'cold-point' here? The coldest temperatures in the geographic distribution between 370 K and 100 hPa? The vertical cold point tropopause?

**3  Technical suggestions**

**page 1**
**l.24:** recommend changing this to 'Due to the warmer dehydration temperatures, anomalously moist air enters...'
**l.25:** typo: 'frank' → 'flank'

**page 2**
**l.13:** 'moist' → 'moisture'
**l.18:** recommend moving 'controlling transport' to after 'processes' instead of before
**l.24:** 'ascent' → 'ascends'

**page 3**
**l.1:** recommend moving 'over the Bay of Bengal and Southeast Asia' to after 'direct convective injection' for readability.

**page 4**
**l.25:** recommend deleting 'It is featured with' for clarity

**page 5**
**l.7:** 'as significant increase' → 'as significant of an increase'

**page 6**
**l.20:** 'described in Section 2 to the data' → 'to the data as described in Section 2'
**l.28:** typo: 'principle' → 'principal'

**page 7**
**l.3:** typo: 'dehydration' → 'dehydrated'

**page 8**
**l.3:** typo: missing 'mid' or 'late' in 'early-to-summer moistening'?

**page 9**
**l.6:** the meaning of 'convective protrusion' here is not clear — do you mean that convection over these regions is particularly deep relative to other parts of the monsoon domain, that convection is particularly frequent in these regions, or something else?
**l.11:** typo: 'studies' → 'study'
* * *

---

## Author Comment (AC1) · 26 Apr 2016

**Replies to the Comments**

April 26, 2016

We are very grateful to the two reviewers for their detailed comments and suggestions to significantly improve our manuscript.

We have made two substantial changes to the manuscript, which are shown below. All detailed changes and point-to-point answers to the reviewers' comments are detailed below. A revised manuscript with all tracked changes is attached.

**(1) A complete analysis of confidence intervals and significance testing**

Time series in Figures 1 and 3 were updated with standard deviation intervals. We have also added significance test for the differences of diabatic heating between August and June in Figure 4b. The results turn out to be very robust and overall conclusions are not critically impacted.

**(2) Direct comparison of tropopause temperatures on the east and west side**

We have also added a direct comparison of tropopause temperature on the east and west side of the Asian monsoon to Figure 3b. The domains for the two sides are shown in Figure 2.

**Reply to anonymous Referee #1 (acp-2016-21-RC1)**

**General comments:**

This paper discussed the contribution of the geographic variation of convection and the associated geographic variation of dehydration locations to the water vapor over Asian monsoon region in the lower stratosphere during boreal summer. The trajectory model simulation provided clear proof of the east-to-west shift of the dehydration location at the intra-seasonal time scale. Further SVD analysis confirms the connection between the convection pattern and the water vapor anomalies. The main concern from my perspective is the statements in many places of the paper like 'warmer tropopause temperature in the west of Asian monsoon region', e.g. Page 1 line 20-24, Page 6 line 16-17 and Page 8 Line 25-26. It seems to me that the author is comparing the tropopause temperature in the west side to that in the east side. However, it is not clear in which region, which latitude or which period it is compared. The author divided the east side and west side of Asian monsoon region by 80-90_E according to the caption of Figure 3. However, from the tropopause temperature shown in Figure 2, the differences of the tropopause temperature between the east side and the west side are not significant. And the author also pointed out that the convection increases over the west side of Asian monsoon region which increases the local diabatic heating. From my understanding, the anomalous convection over the west Asian monsoon region should lead to stronger upwelling and relatively cold temperature in the tropopause layer, which controls the dehydration. Actually, the pattern of dehydration location and of the tropopause temperature is associated with each other. That means when the convection increases over western side of Asian monsoon region, the cold point temperature should correspondingly decreases to some degree. I suggest author gives a direct comparison of the tropopause temperature of east side and west side to clarify this point.

Furthermore, Figure 1 shows the variation of tropopause temperature is not able to fully explain the variation of seasonal water vapor variation, which is the motivation of the work.

I have some questions concerning the domain and the magnitude of interannual variabilities, which are specified in the following part.

The paper describes interesting result, which contributes to complete the picture of moisture center over Asian summer monsoon region. Overall, the paper is nicely structured and presented. I suggest it is published after clarifying the questions above.

**Reply:**

Thanks for those helpful comments.

We've added a direct comparison of the weighted tropopause temperature of east side and west side to the revised manuscript, see Figure 3b in the revised manuscript. In order to avoid the artificial effects of the domain selections, we calculated the average tropopause temperatures in the two domains by taking dehydration frequencies into consideration as weights. We agree with the reviewer that when the convection increases over the west side of the Asian monsoon region, the cold point temperatures could correspondingly decrease by some degrees. But overall, the temperatures over the west side are higher than those over the east side. In addition, the magnitude of temperature differences is larger than the sub-seasonal variations (see Figure 3b). Therefore, an increase of dehydration frequencies over the west side of the Asian monsoon region would increase the fraction of air parcels dehydrated at relatively warmer temperatures. And this positive impact is significantly larger than the potential offset influence of decreased tropopause temperature induced by increased convection.

**Specific comments:**

• Pg. 2, line 21: A recent paper (Ploeger et al. 2015, ACP) intensively discussed the variability of a PV-based transport barrier of Asian monsoon anticyclone. This study is highly related here and I recommend to cite this study. doi:10.5194/acp-15-13145-2015

**Reply:**

Yes, we have added the citation to Pg. 2, line 26.

• Pg. 3 , line 28: I suppose the OLR data is also daily anomalies according to the section 3.2. So add 'of' after 'water vapor and' in order to avoid the misunderstanding.

**Reply:**

Done, see Pg. 4, line 10.

• Pg. 4, line 29: Is the weighting functions the weighting matrix of MLS averaging kernels? If yes, please clarify here.

**Reply:**

Yes, we used the weighting matrix of the MLS averaging kernals and we have clarified this in Pg. 5, line 2.

• Figure 1: First, this figure shows the 9-year climatology of water vapor and tropopause temperature. The intra-seasonal variations are usually can be 'offset' by averaging over several years. Can you comment on how large is the interannual variability of temperature? Does this strong intra-seasonal variations of temperature attribute to some particular year or is it a common feature for this domain during boreal summer? Perhaps it worth to add the standard deviation to the tropopause temperature.

**Reply:**

We have added standard deviation intervals for the tropopause temperature in Figure 1. We have checked individual years; such an intra-seasonal variation of temperatures is a common phenomenon during the period 2005-2013. We are not sure what caused such an intra-seasonal feature, which may be linked to some intraseasonal oscillation features in the Asian monsoon region (e.g., Lau and Chan, 1986; Kikuchi et al., 2012).

• Figure 1: Second, you mention that the same domain used for area-averaging the tropopause temperatures as R15, which is 15-32_N, 70-120_E. However, I checked R15 and the domain 15-30_N, 70-120_E is used. Besides, you also use 15-30_N in Figure 4 of averaging the diabatic heating rate. From Figure 2, it is seen that the gradient of tropopause temperature around 30_N is very large and the 2 degrees could influence the variation of temperature shown in this plot. I suggest to show the tropopause temperature averaged over 15-30_N, 70-120_E. Otherwise the author could compare the results between the 2 domains and clarify the results stays the same.
    **Reply:**
    Thanks for pointing this out. 32ᵒN was a typo and it has been corrected.

• Figure 2 and 3: I suggest to add boxes in Figure 2 to show the domains of west side and east side mentioned in the caption of Figure 3.
    **Reply:**
    Thanks for this great suggestion. We have added two boxes in Figure 2 to indicate the domains of west side and east side of the Asian monsoon region.

• Figure 5: the subfigures are too small. It is better to enlarge the figure, especially those color bars.
    **Reply:**
    Thanks for this suggestion. We have rearranged the order of Figure 5 to make figures bigger.

• Pg. 8, line 10: it should be '(Fig.1 and Fig.6a-b)'
    **Reply:**
    Changed.

**Reply to anonymous Referee #2 (acp-2016-21-RC2)**

First, my apologies to the authors and the editor for the long delay in publishing this review.

This manuscript examines how changes in the distribution of convective sources and dehydration locations of air in the Asian monsoon upper tropospheric anticyclone affect the amount of water vapour entering the lower stratosphere in this region. The concept is worthwhile, and the paper makes some valid points about how seasonal and intraseasonal variability in convective sources influence the moisture content of air near the tropopause. However, some aspects of the methodology and argument are problematic, and require more justification at the very least.

The biggest weakness of this paper is that it takes water vapour variations at 100 hPa to be representative of the 'lower stratosphere', neglecting previous work indicating that final dehydration for air entering the tropical stratosphere via this region typically occurs at lower pressures / higher altitudes. This does not necessarily invalidate the core conclusions of this paper (the processes controlling water vapour variability at 100 hPa are also important to understand, particularly if they propagate to higher levels), but it does imply strong limitations on their applicability that are not effectively communicated or explored in the paper. At the very least, the authors should clarify that 'LS' in this case means 100 hPa, and discuss the limitations that that entails. Even better, the authors could use their trajectory simulations to connect the results and conclusions at 100 hPa to final dehydration statistics and stratospheric entry mixing ratio. In other words: do these intraseasonal differences in source location / temperature distribution / transport affect the amount of water vapour entering the global stratosphere via the Asian monsoon anticyclone, and, if so, how much? These additions would help tremendously in establishing how this work fits in the context of other studies of water vapour transport and variability in this region.

**Reply:**

Thanks very much for these very helpful comments and suggestions. We have clarified in the manuscript that we use 100 hPa to represent lower stratosphere in Page 5, Line 12 in the beginning of the Results Section.

Although our focus is on the monsoon water vapor at 100 hPa, we have also examined the corresponding results at higher levels. Results at 82 hPa are almost the same as the 100 hPa. This was mentioned at Page 8, Line 8 in the original manuscript.

We mainly focus on the LS, particularly at 100 hPa and 82 hPa, because we could typically observe the moisture center over the Asian monsoon region during summer at 100 hPa and 82 hPa, with much a more weakened phenomenon at 68 hPa and even disappeared at higher levels based on Aura MLS observations. The variability is much smaller above 68 hPa than lower levels. In previous papers (e.g., Dessler et al., 2013; Randel et al., 2015), they usually use 100 hPa or 82 hPa to represent the LS. Our use of 100 hPa was influenced by the study of Randel et al. (2015) that used 100 hPa to represent the LS.

Regarding the final dehydration height, a recent paper using the same trajectory model (Dessler et al., 2016) states that "Dehydration events at altitudes above 93 hPa do occur, but they remove relatively small amounts of water: the water vapor mixing ratio at 79 hPa is within a few percent of the value at 93 hPa." Therefore, water vapor variations at 100 hPa and 82 hPa are representatives of water vapor variations in the 'LS'.

In the study of Dessler and Sherwood (2004), they calculated stratospheric entry mixing ratio of water vapor in the trajectory simulations by averaging the H2O mixing ratio of parcels between 75 and 91 hPa (16.8–18.5 km), similar to the MLS 82 hPa weighting function. Therefore, they assumed ∼82 hPa water vapor values in the trajectory model to be the entry values into the stratosphere. Since **our results are very robust for 82 hPa**, we believe that the influence of geographic dehydration variations could also impact the stratospheric entry mixing ratio in a similar way, especially over the Asian monsoon region, i.e., the entry values for western trajectories would be significantly higher than those with convective sources over the eastern side of the region. While the amount of LS water vapor in the Asian monsoon region is our focus, further studies are still needed to investigate the influence of Asian monsoon water vapor abundance (or entry mixing ratio) on global LS water vapor. We appreciate that you pointed out the limitations. The influence of the seasonal and intraseasonal differences in convective and dehydration center on the amount of water vapor transport to the global stratosphere is a future study.

**1 General comments**

1. My main concern is that the analysis focuses almost exclusively on water vapour at 100 hPa, and particularly that this is assumed to represent lower stratospheric water vapour. The vertical location of dehydration for air entering the stratosphere varies quite a lot, and is typically higher (in altitude) than 100 hPa. Would the results still be valid for variations in water vapour at 83 or 68 hPa, or are they only relevant to a shallow layer bracketing the tropopause? If I have understood the analysis correctly, this might be checked by analyzing the 'final dehydration' locations and temperatures for these trajectories during transit to the stratosphere in addition to the 'latest dehydration' locations and temperatures for the model results at 100 hPa. Are the statistics of final dehydration for these trajectories significantly different from those with convective sources over the eastern side of the region? Regardless, more needs to be done here, either to connect these results more clearly to stratospheric entry mixing ratio (post-final dehydration) or clearly distinguish between studies for which 'LS' means 'above the tropopause layer' and/or 'after final dehydration' and this study (where 'LS' means 100 hPa, well within the tropopause layer and likely prior to final dehydration).

**Reply:**

As mentioned above, the results are still valid for water vapor variations at 82 hPa. And we also found that most of the relatively wet parcels at 68 hPa are from the dehydration over the

west side with warmer tropopause temperatures. The reasons why we use 100 hPa and 82 hPa to represent LS in the Asian monsoon region can be found in the above reply.

The concept to use trajectory model to look at dehydration locations in this study is different from previous studies (Schoeberl et al., 2012; Wang et al., 2015), in which, they define final dehydration points as where parcels underwent final dehydration and stayed at altitudes higher (pressure lower) than 90 hPa for at least 6 months since the last time they were dehydrated (FDP). This guarantees that parcels already crossed the cold-point tropopause (~380 K or ~100-94 hPa) and experienced their final dehydration (Wang et al., 2015). In this way, the greatest FDP frequency would mostly occur at locations with extremely low temperatures (eg., tropical cold-point tropopause layer), as the trajectory model only records the final dehydration points at the long paths. Besides, in reality, for example, most of the air parcels at 100 hPa (or 82 hPa) haven't gone through the final dehydration presented in the above studies. Thus, in order to study observed water vapor variations at those levels, most recent dehydration (MRD) instead of FDP statistics may be more applicable. In our paper, we select all the air parcels at 100 hPa (or 82 hPa), and find out the MRD statistics that determine the amount of water vapor in each air parcel. This should be the correct way to study what really controls the variations of water vapor at a particular level in details.

2. I'm not convinced that the idealized experiments that separate the dehydration temperatures and dehydration locations are viable in this case. This approach works well when either temperature changes or circulation changes are dominant (and therefore separable), but has little meaning when temperature and circulation changes are tightly coupled. Another way of thinking about this is that separability is a justifiable assumption in situations where changes in dehydration location are dominated either by (1) an unchanged circulation sampling a modified temperature distribution or (2) a modified circulation sampling an unchanged temperature distribution. Too much overlap between these situations results in degeneracy, at which point the contributions of temperature changes and circulation changes cannot be reliably distinguished. My expectation is that in this case the tight couplings among convection, circulation (especially diabatic heating) and temperature at 100 hPa violate separability, as also briefly mentioned by reviewer #1. I am willing to be convinced otherwise, but additional justification for these simulations is needed if they are to be used as supporting evidence here.

**Reply:**
The idealized experiments were designed to compare the relative impact of dehydration locations and temperature changes on water vapor increase from June to August. The concept was very simple, i.e., to look at how the westward shift of dehydration locations changes the water vapor entering the LS on seasonal scale. We agree that convection, circulation and temperature are tightly coupled. The experiments were not designed to separate the influence of convection, circulation and temperature. The concept is especially useful when one wants to assess the stratospheric water vapor changes from the large-scale tropopause temperature changes, while neglecting the changes of dehydration locations. We have added one sentence to Pg. 7, Line 11, "*These idealized experiments indicate that we may underestimate the water vapor variations solely based on the large-scale temperature changes without considering the changes of dehydration statistics associated with the large-scale circulation changes.*"

3. This work would benefit from a more complete analysis of confidence intervals and significance testing. As also noted by reviewer #1, many of the arguments rely on changes and/or differences that are relatively small. This is particularly relevant for the time series in Fig. 3 and the August minus June differences shown in Fig. 4 and Table 1, and otherwise reported in the text.

**Reply:**
Yes. We have added the standard deviation intervals and significance tests for those changes and differences to Figures 1-4.

4. The text is clear for the most part, but the manuscript would benefit from English-language editing by a colleague or professional editor. There are a few points where editing will be necessary to improve the clarity; see technical comments below.

**Reply:**

Thanks for this suggestion. We have edited the languages in this paper again according to your suggestion.

2 Specific comments p.2, l.24-25: this sentence is vague and should be reworded for clarity — at seasonal time scales and large spatial scales both the temporal evolution and geographic distribution of LS water vapour are correlated with convective activity, it's just that these correlations do not generally extend to variability within the anticyclone itself.

**Reply:**

We have rephrased the sentence. Please see Pg.3, Line 2.

p.2, l.28: Given the uncertainties and competing hypotheses put forward by subsequent studies, it would be more appropriate to change 'they can' to 'they proposed that this convection can'

**Reply:**

Changed. Please see Pg. 3, Line 6.

p.3, l.6: Wright et al. (2011) did find large discrepancies among the different reanalysis data sets, but the qualitative results were robust: trajectories originating from convection over Tibet were consistently moister but less numerous than trajectories originating from convection over the other regions, so that these trajectories had relatively limited impacts on water vapour in the global tropical LS.

**Reply:**

We have modified the sentence. Please see Pg. 3, Line 14.

p.3, l.23: It would be useful to note also the relative precision here (as xx–yy%)

**Reply:**

We have added percentage precisions to Pg. 4, Line 6.

p.4, l.16: In this case, since the focus is on the evolution at 100 hPa, I presume that 'latest dehydration' refers to most recent dehydration rather than final dehydration. This choice should be stated explicitly to prevent confusion – 'latest dehydration' is by itself too vague, as it could mean either 'most recent' or 'final'.

**Reply:**

We have changed "latest dehydration" to "most recent dehydration (MRD)" in the revised version to avoid confusion.

p.4, l.18-19: How is this done, by gridding the simulated water vapour mixing ratios and then applying the averaging kernels to construct a vertical grid? I recommend expanding slightly on this description. Also, as noted later on this page, the exclusion of simulated values below 100 hPa results in a dry bias. I assume this statement is based on testing the sensitivity to whether those values are included. Does this testing indicate whether including/excluding the simulated values at lower levels has any impact on the qualitative evolution of the variability?

**Reply:**

Yes, we first gridded the simulated water vapor mixing ratios and then applied the MLS averaging kernels to the gridded water vapor fields to construct the same vertical grids as the Aura MLS data. We have expanded the description, please see to Pg. 5, Line 3.

Yes, excluding the simulated values at lower levels does have impact on the water vapor variability. We have added a note to Pg. 8, Line 27.

p.5, l.5-9: It is very difficult for the reader to evaluate the statement that '100 hPa temperatures over the southeastern flank of the anticyclone ... do not show as significant increase as water vapour from May and June to August'. This argument should be made more quantitative. This could be as simple as a calculation relating the May/June to August mean temperature change to a fractional change in mean saturation specific humidity (with appropriate uncertainty estimates), which can then be compared to the fractional change in simulated water vapour mixing ratio at 100 hPa (with appropriate uncertainty estimates).

**Reply:**

We have rephrased the sentence to make it more accurate, please see Pg. 5, Line 27. And we have added Figure 2e to compare with the time series of temperatures shown in Figure 1.

p.5, l.13: (Fig. 2) Does 'tropopause temperatures' mean '100 hPa temperatures', or are these evaluated at a diagnosed tropopause?

**Reply:**

The tropopause temperatures were calculated from ERA-Interim temperature fields based on the WMO tropopause definition.

p.5, l.20: (Fig. 3) Here it would be helpful to include also the evolution of mean 'latest dehydration' temperatures over the eastern and western parts of the domain, with uncertainty estimates. This would help to clarify that it is in fact the shift in dehydration location (and not the temperature evolution) that dominates the seasonal evolution of water vapour at 100 hPa, and could perhaps supplement or replace the idealized simulations in the overall argument.

**Reply:**

Yes, we have added the evolution of weighted tropopause temperatures with uncertainty intervals over the east and west side of the Asian monsoon region to Figure 3. This would supplement the idealized simulations with further clarifications.

p.5, l.26: (Table 1) If using these simulations, it would be useful also to include the August–August results to give a quantitative benchmark for evaluating the idealized June–August and August–June simulations. I know that these are shown in Fig. 1, but so are the June–June results. I could not find this number reported anywhere in the text.

**Reply:**

Yes, we now have included the August-August results in the revised manuscript.

p.6, l.12: (Fig. 4) Is there any benefit to including the profiles of diabatic heating below 300 hPa? Including these estimates requires the use of a relatively large scale, and makes it difficult to distinguish the variations in the UTLS (which is what we are particularly interested in). Moreover, there is a negative anomaly centered around 70 hPa above the location with enhanced convective activity in August relative to June. Is this negative anomaly significant? If so, this suggests that upward motion in this region is weaker than during June above 100 hPa, which might mean that the trajectories involved circuit the anticyclone more times during ascent. This relates to general comment #1 above: how much does this westward shift of convective source location ultimately impact stratospheric entry mixing ratios?

**Reply:**

One benefit to include the profiles of diabatic heating below 300 hPa is that we could look at the large-scale convection changes in the whole column of the troposphere. The negative anomaly at 70 hPa is significant, please see the significance test in the revised Figure 4. This indeed would slow down the ascending motion above 100 hPa a little bit. It also involves tight

coupling with temperature fields and anticyclonic circulation. We don't know how this convection induced diabatic heating changes in the LS would influence stratospheric entry mixing ratio. We decide to leave this interesting topic for future studies.

p.6, l.20: (Fig. 5) The use of colours here is confusing, with red sometimes meaning a positive change and sometimes a negative change. Moreover, 'physically consistent with a moist anomaly in the UT' is sometimes indicated by the yellow/orange/red half of the colour scale and sometimes by the green/blue/purple half of the color scale. I recommend that you either make the use of this default colour table logically consistent across panels or use different colour tables for anomalies in different quantities. I also agree with reviewer #1 that the fonts are too small and difficult to read in several of the panels included in Fig. 5, and I can barely even see the variations of the lines in panel (f). If the variations for all of the years are necessary, perhaps it would be better to make panel (f) a separate figure and split it into multiple panels, one for each year? If not, it might be best to show the variations for a selected time period covering one or two years, so that those variations are easier to identify in the figure.

**Reply:**

In the revised manuscript, we have reordered the figures in Figure 5. Moreover, we change the color bars to indicate red (blue) color as increase (decrease). Figure 5f has also been enlarged to have a clear view of the intraseasonal variation.

p.7, l.1: Which data is used to identify the increase in cirrus clouds?

**Reply:**

Cloud-Aerosol Lidar and Infrared Pathfinder Satellite Observations (CALIPSO) 2006-2013 daily observations was used to identify the increase of cirrus clouds. I have added this information to the revised paper. Please see Pg. 8, Line 7.

p.7, l.3: What mechanism drives the enhanced ascending motion? Enhanced latent heating above 370 K? Enhanced cloud radiative heating? Is this enhanced ascending motion consistent between ERA-Interim and MERRA? This could be explored by looking at the components of the heat budget — both ERA-Interim and MERRA provide clear-sky and all-sky radiative heating products.

**Reply:**

Most of the enhanced latent heating above 370 K and below 380 K is still due to latent heating. While beyond the 380 K, the radiative heating dominates. We will not discuss too much here as we will have a separate paper to be submitted systematically talking about the relationships between convection and diabatic heating in the atmosphere. Yes, the enhanced ascending motion is consistent between ERA-Interim and MERRA.

p.7, l.9: (Fig. 6) For clarity, the definitions of 'wet' and 'dry' days should perhaps be moved from l.20 to here.

**Reply:**

Done.

p.7, l.18-19: It would be useful to include the correlation for traj_MERRA if data from the mismatched period in 2006 is excluded.

**Reply:**

The correlation for traj_MERRA is calculated by including all the period between 2004-2013, same as traj_ERAi. 2006 is not excluded here, that's why the correlation is lower than traj_ERAi.

p.8, l.25: The presented work only supports this statement if we consider 100 hPa to be representative of the LS in this region — no confirmation has been shown that this seasonal

evolution in the convective source extends to lower pressures / higher altitudes, which should also be considered part of the LS.

**Reply:**

Please see the reply to comment #1.

p.9, l.4: What is meant by 'cold-point' here? The coldest temperatures in the geographic distribution between 370 K and 100 hPa? The vertical cold point tropopause?

**Reply:**

We have changed it to "cold-point tropopause temperatures over southeast Asia".

**3 Technical suggestions**

page 1 l.24: recommend changing this to 'Due to the warmer dehydration temperatures, anomalously moist air enters...' l.25: typo: 'frank' ! 'flank'

**Reply:**

Changed.

page 2 l.13: 'moist' ! 'moisture' l.18: recommend moving 'controlling transport' to after 'processes' instead of before l.24: 'ascent' ! 'ascends'

**Reply:**

Changed.

page 3 l.1: recommend moving 'over the Bay of Bengal and Southeast Asia' to after 'direct convective injection' for readability. page 4 l.25: recommend deleting 'It is featured with' for clarity

**Reply:**

Changed.

page 5 l.7: 'as significant increase' ! 'as significant of an increase'

**Reply:**

Changed to "increase concurrently with" to make the statement more accurate. Please see Pg. 5, Line 27.

page 6 l.20: 'described in Section 2 to the data' ! 'to the data as described in Section 2' l.28: typo: 'principle' ! 'principal'

**Reply:**

Changed.

page 7 l.3: typo: 'dehydration' ! 'dehydrated' page 8 l.3: typo: missing 'mid' or 'late' in 'early-to-summer moistening'?

**Reply:**

Changed.

page 9 l.6: the meaning of 'convective protrusion' here is not clear — do you mean that convection over these regions is particularly deep relative to other parts of the monsoon domain, that convection is particularly frequent in these regions, or something else?

**Reply:**

Yes, we have changed the "convective protrusion" to "frequent and deep convective protrusions" to make it more clear.

l.11: typo: 'studies' ! 'study'

**Reply:**

**Reference**

Dessler, A., Ye, H., Wang, T., Schoeberl, M., Oman, L., Douglass, A., Butler, A., Rosenlof, K., Davis, S., and Portmann, R.: Transport of ice into the stratosphere and the humidification of the stratosphere over the 21st century, Geophys. Res. Lett., 2016.

[revised manuscript text omitted]